# GITCO: Gated Inference-Time Context Optimization in TSFMs

**Manya Pandey** [1 2]  **Dhruv Kumar** [1 3]  **Murari Mandal** [1 2 †]  **Saurabh Deshpande** [1 †]

## Abstract

Patch-based Time Series Foundation Models (TSFMs) suffer from *context poisoning:* structurally anomalous patches capture disproportionate attention and silently degrade zero-shot forecast quality. We propose improving TSFM accuracy at inference time by optimizing the input context rather than modifying model weights. We present **GITCO** (Gated Inference-Time Context Optimization), a lightweight three-component framework: *Gate, Router,* and *Critic* that selectively identifies and suppresses harmful patches without any parameter updates. Evaluated on TimesFM 2.5 across 53 GIFT-Eval datasets under K-fold cross-validation, GITCO achieves an average +1.95% MASE reduction on TimesFM 2.5 while capturing 89.9% of the improvement upper bound. We introduce *context sensitivity profiles* as a new characterizable property of TSFMs: the mapping from time series meta-features to expected accuracy improvement under inference-time context intervention, shaped jointly by model architecture and the statistical structure of the data.

## 1. Introduction

The dominant paradigm for improving Time Series Foundation Models (TSFMs) targets the model itself: larger pretraining corpora, architectural modifications, and task-specific fine-tuning (Liang et al., 2024). Recent work (Hua et al., 2026) has begun to explore inference-time scaling as an alternative, leveraging additional test-time computation via multi-sample and diversified decoding to improve

performance without updating parameters. In production deployments where weights are frozen and compute is limited, these strategies remain challenging to apply. We build on the paradigm of input-centric inference-time optimization, exploring its systematic application to TSFMs.

**The Failure Mode.** Patch-based TSFMs such as TimesFM 2.5 (Das et al., 2023) and Chronos2 (Ansari et al., 2024) tokenize a fixed-length context window into patches (Nie et al., 2023) and forecast via attention over this sequence. When any patch contains a misleading signal, a volatility burst, a level shift, a spurious seasonal artifact, it can capture disproportionate attention weight and corrupt the forecast even when surrounding context is clean. We term this *context poisoning*. The vulnerability of attention-based forecasting to such signals is consistent with broader critiques of transformer architectures in time series modeling (Zeng et al., 2022). It is not rare: across 53 GIFT-Eval datasets, over 50% of series show marginal improvability and 22 datasets benefit meaningfully from targeted patch intervention. Critically, improvable datasets are *predictable* from cheap, model-agnostic meta-features (Hyndman et al., 2023) computed on the input and that predictability is the foundation of our approach.

**Our Approach.** Analogous to how chain-of-thought (Wei et al., 2022) and self-consistency (Wang et al., 2023) improve LLM outputs by refining the reasoning context rather than modifying weights, consistent with the shift toward test-time compute optimization (Snell et al., 2024): we propose refining the *input context* of frozen TSFMs at inference time. We present **GITCO** (Gated Inference-Time Context Optimization): a **Gate** decides whether to intervene; a **Router** selects among three expert probe Critics; a **Critic** identifies the most disruptive patch, which is then smoothed via SMA ($w{=}5$).

**Contributions.** We propose **GITCO**, the first inference-time context optimization framework for TSFMs, evaluated on two frozen architectures across 53 GIFT-Eval (Aksu et al., 2024) datasets under $K$-fold cross-validation. We introduce **context sensitivity profiles** ($\Phi_M$). Empirical evidence shows that context improvability differs across models not only in decision boundary shape but in intrinsic learnability from a shared meta-feature vocabulary, establishing $\Phi_M$ as

† Equal supervision. Code available at https://github.com/birla-ai-labs/gitco. [1]Birla AI Labs, Mumbai, India [2]KIIT, Bhubaneswar, India [3]BITS Pilani, Pilani, India. Correspondence to: Saurabh Deshpande <saurabh.deshpande-c@oab.adityabirla.com>, Murari Mandal <murari.mandal-c@oab.adityabirla.com>.

*ICML 2026 Workshop on Foundation Models for Structured Data*, Seoul, South Korea. Copyright 2026 by the author(s)

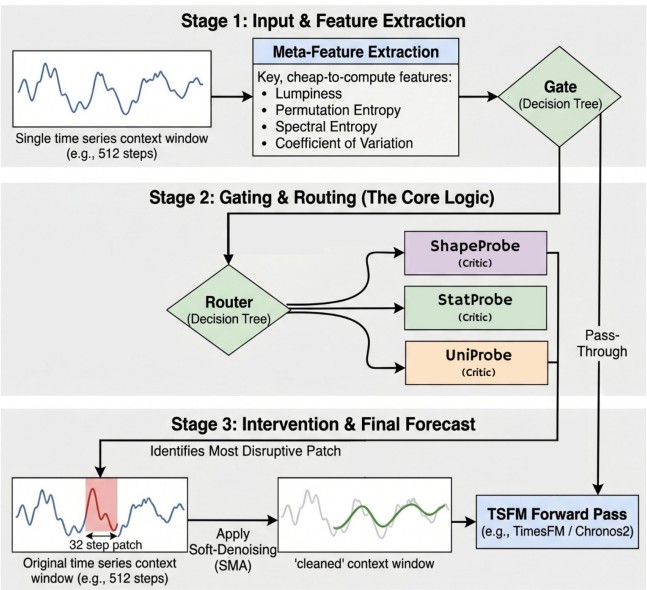

*Figure 1.* **The GITCO Pipeline**: A raw 512-step context window (16 patches of 32 steps) is passed to the Gate, which computes meta-features to decide whether intervention is warranted. If so, the Critic scores all 16 patches and identifies the most disruptive one (highlighted red); soft-denoising via SMA produces the GITCO-conditioned context.

a model-specific, vocabulary-dependent property.

## 2. Related Work

Time Series Foundation Models (TSFMs) achieve strong zero-shot forecasting via large-scale pretraining on diverse corpora (Liang et al., 2024; Das et al., 2023; Ansari et al., 2024), with most progress driven by training-time scaling (data, architectures, adaptation). In contrast, inference-time optimization remains underexplored in time series, despite its success in NLP via test-time compute scaling (Sun et al., 2020; Snell et al., 2024). The only closely related work (Hua et al., 2026) improves forecasting through multi-sample diversified inference using input perturbations and aggregation. Prior TSFM methods such as in-context fine-tuning (Faw et al., 2024) and architecture-driven in-context learners (Auer et al., 2025), or internal hidden-state interventions (Sanyal et al., 2025) focus on prompt design (Gruver et al., 2024), model architecture, or latent manipulation. GITCO differs by refining the input context itself: a lightweight Gate-Critic-Router pipeline suppresses misleading patches within a single history, enabling model-agnostic gains on frozen TSFMs with minimal overhead.

## 3. GITCO

GITCO operates as a modular, three-stage inference-time wrapper around any frozen patch-based TSFM. Given an input context $X \in \mathbb{R}^{N \times P}$ consisting of $N$ patches of length $P$, the system first decides *whether* to intervene (Gate), then *how* to intervene (Router), and finally *where* to intervene (Critic). If the Gate abstains, the baseline TSFM forecast is returned unmodified. The full pipeline is summarized in Algorithm 1.

---

**Algorithm 1** GITCO Inference

---

**Require:** Context $X \in \mathbb{R}^{N \times P}$, meta-features $\phi(X)$
**Ensure:** Forecast $\hat{Y}$
1: $g \leftarrow \text{Gate}(\phi(X)), \quad g \in \{0, 1\}$
2: **if** $g = 0$ **then**
3:     **return** $\hat{Y} \leftarrow \text{TSFM}(X)$
4: **end if**
5: probe $\leftarrow$ Router$(\phi(X))$, probe $\in$ $\{\texttt{ShapeProbe}, \texttt{StatProbe}, \texttt{UniProbe}\}$
6: $i^* \leftarrow \arg\max_{i \in \{1, \ldots, N\}} \text{Critic}_{\text{probe}}(X)$
7: $X' \leftarrow \text{SMA}_5(X, i^*)$
8: **return** $\hat{Y} \leftarrow \text{TSFM}(X')$

---

### 3.1. The Gate

The Gate is a binary classifier $g : \mathbb{R}^d \rightarrow \{0, 1\}$ mapping input meta-features $\phi(X)$ to an intervention decision. Its design is governed by an explicit asymmetric loss, prioritizing precision over recall. This reflects the empirical reality that false positives (disrupting a clean signal) degrade accuracy by a magnitude $|\mu_-|$ that exceeds the mean gain $\mu_+$ from true positives. By enforcing $|\mu_-| > \mu_+$, the Gate mitigates destructive interventions on stable sequences.

Under $K$-fold cross-validation, the TimesFM 2.5 Gate converges on a robust decision boundary driven by frequency-domain characteristics. The primary discriminant is seasonality strength, refined by spectral entropy. The induced logic is non-monotonic: it triggers intervention on signals lacking clear seasonality or exhibiting high spectral entropy, while correctly abstaining on intermediate, structured sequences. This reliance on spectral properties, rather than simple volatility metrics, identifies a learnable signature for TimesFM's specific failure modes, showing its unique context sensitivity profile.

### 3.2. The Router

Conditional on a Gate intervention decision ($g = 1$), the Router maps meta-features $\phi(X)$ to an expert probe selection $r \in \{\texttt{ShapeProbe}, \texttt{StatProbe}, \texttt{UniProbe}\}$. For TimesFM 2.5, this logic bifurcates along a volatility axis: stable series (low coefficient of variation) are routed to $\texttt{UniProbe}$ or, if outlier density is high, to $\texttt{StatProbe}$. Highly volatile series are predominantly routed to the CNN-based $\texttt{ShapeProbe}$, effectively isolating persistent random walks (Hurst exponent near 1.0) where local geometric

anomalies mimic structural shifts.

While the TimesFM Gate relies on frequency-domain features (seasonality, spectral entropy) to determine *whether* to intervene, the Router relies on magnitude and persistence features to determine *how* to intervene, revealing an orthogonal decomposition of contextual vulnerabilities. Although 11-fold cross-validation accuracy on the 3-class routing problem is low ($33.3\% \pm 28.4\%$), the Router functions effectively as a *regret minimizer*. Because the improvement landscape is relatively flat and the second-best probe frequently offers comparable gains, precise probe selection is not critical; this dynamic allocation strategy successfully captures $89.9\%$ of the theoretically achievable oracle improvement.

### 3.3. The Critic

To ensure cross-architecture comparability, GITCO employs a shared vocabulary of three expert Critics (`ShapeProbe`, `StatProbe`, and `UniProbe`) across all evaluated TSFMs. Given a context window $X = \{p_1, \ldots, p_N\}$, the selected Critic acts as a relative ranker, utilizing a lightweight MLP to assign a disruption probability $c_i \in [0, 1]$ to each patch. The system isolates the maximally confounding patch, $\hat{i} = \arg\max_i c_i$, and applies localized soft-denoising via a 5-point Simple Moving Average, yielding a refined context $X' = \text{SMA}_5(X, \hat{i})$. We employ SMA as an efficient heuristic to suppress high-frequency, non-structural anomalies; preliminary ablations confirm that downstream forecast improvements are dominated by the precise spatial localization of $\hat{i}$, rather than the complexity of the filtering operator itself.

## 4. Experiments and Results

We present our evaluation in three parts: the out-of-sample end-to-end performance of the pipeline on TimesFM 2.5 (Section 4.2), the discovery of architectural conditioning through the learnability asymmetry observed in Chronos2 (Section 4.3), and an ablation study confirming the symbiotic necessity of the Gate and Router components (Section 4.4).

### 4.1. Experimental Setup

We evaluated 53 diverse GIFT-Eval datasets (Aksu et al., 2024) (Appendix C.1). Stable MASE estimates for Gate and Router oracle labels were generated via a sliding forecast window (Appendix C.2). To preclude data leakage, all results reflect out-of-sample predictions from rigorous $K = 11$-fold cross-validation, relying solely on input-derived meta-features $\phi(X)$. Our primary metrics (Table 1) include Gate precision and recall, Captured Improvement Ratio (CIR; fraction of attainable gain captured), and aggre-

*Table 1.* End-to-end pipeline performance on TimesFM 2.5. Results are out-of-sample under $K = 11$ cross-validation across $N = 53$ datasets. The precision-biased gate is designed to avoid baseline degradation.

| Metric | Value |
|---|---|
| GATING DIAGNOSTICS | |
|     Gate Precision | 78.0% |
|     Gate Recall | 57.6% |
|     Datasets Intervened | 24 / 53 |
| PIPELINE EFFICACY | |
|     Captured Improvement Ratio (CIR) | **0.899** |
|     Total MASE Improvement ($\sum \Delta$MASE) | **+1.032** |
|     Mean MASE Reduction (all datasets) | **+1.95%** |
|     Mean MASE Reduction (intervened, $n = 24$) | **+4.30%** |

gate MASE reductions (total $\sum \Delta$MASE and mean %). All comparisons target the frozen, zero-shot baseline.

### 4.2. TimesFM 2.5: High Value Capture via Precision Gating

Evaluated under strict $K = 11$-fold cross-validation across 53 datasets, the GITCO pipeline demonstrates consistent end-to-end efficacy on TimesFM 2.5 (Table 1). Intervening on 24 of 53 datasets, the pipeline delivers a mean MASE reduction of $+1.95\%$ across all 53 datasets ($\sum \Delta$MASE $= +1.03$ absolute units), with a mean improvement of $+4.30\%$ restricted to the intervened subset. The **Captured Improvement Ratio (CIR)** of $0.899$ indicates that the pipeline recovers nearly $90\%$ of the improvement achievable by a hindsight-optimal intervention policy under the defined probe vocabulary and denoising operator, despite operating entirely from input meta-features at inference time.

This disproportionate value capture follows directly from the Gate's precision-first operating point ($78.0\%$ precision, $57.6\%$ recall). Out-of-sample analysis reveals a steep asymmetric penalty: $83.3\%$ of false-positive activations directly degrade baseline accuracy, motivating the Gate's conservative decision boundary. By sacrificing recall to guarantee safety, the pipeline reliably isolates improvable contexts while avoiding destructive overrides on clean series. Standard classification accuracy significantly underestimates the pipeline's practical value; it functions as a regret minimizer, not a classifier.

### 4.3. Chronos2: The Learnability Asymmetry

Applying the identical $K = 11$-fold gate induction protocol to Chronos2 using the same meta-feature vocabulary produced no deployable classifier. All induced boundaries exhibited low precision and recall across folds, with no feature or combination yielding stable split points. This is a

*Table 2.* Component ablation on TimesFM 2.5. $\Sigma\Delta\%$ is the sum of per-dataset percentage MASE reductions across all intervened datasets; the full GITCO value of $+57.33\%$ corresponds to a mean MASE reduction of $+1.95\%$ averaged across all 53 datasets ($\sum \Delta\text{MASE} = +1.03$ absolute units).

| System Variant | $\Sigma\Delta\%$ | Precision (%) |
|---|---|---|
| Always Intervene | +4.41 | 35.85 |
| Gate Only | +24.83 | 45.83 |
| Router Only | +42.16 | 37.74 |
| **GITCO (Gate + Router)** | **+57.33** | **78.0** |

negative learnability result, not an absence of context poisoning. Oracle analysis confirms the improvement signal is real: SMA-based denoising improves 12–24 of 53 Chronos2 datasets in principle, and the Critics correctly identify which patch to suppress. The signal simply cannot be predicted from the features that characterize TimesFM's failure modes including lumpiness, spectral entropy, and coefficient of variation, which together account for the full TimesFM gate. TimesFM's context sensitivity profile is compact and learnable at $N = 53$; Chronos2's is not from the same vocabulary, the same sample size, and the same induction procedure. $\Phi_{\text{Chronos2}}$ either requires features outside the span of our vocabulary or exhibits a diffuse boundary that resists compact characterization at this scale. Context sensitivity profiles differ across architectures not merely in shape, but in intrinsic learnability.

### 4.4. Component Ablation: Gate and Router Contributions

Table 2 isolates the contribution of each component on TimesFM 2.5. $\Sigma\Delta\%$ aggregates per-dataset percentage improvements across intervened datasets and should be read comparatively across rows rather than as a standalone magnitude; the full GITCO figure of $+57.33\%$ corresponds to the mean MASE reduction of $+1.95\%$ reported in Section 4.2.

Router-Only achieves $\Sigma\Delta\% = +42.16\%$ but at only $37.74\%$ precision: without gating, indiscriminate intervention destroys value on clean series faster than better routing can recover it. Gate-Only recovers safety ($45.83\%$ precision) but leaves substantial improvement unrealized at $+24.83\%$, since a fixed default probe cannot adapt to heterogeneous series characteristics. Always-Intervene is the worst outcome on both metrics, confirming that the denoising operator is not generically beneficial — consistent with the Gating Primacy Principle (§B). Full GITCO achieves the strongest result on both axes, an emergent property of sequential *Gating-then-Routing*: the Gate makes intervention safe; the Router makes safe intervention powerful.

## 5. Conclusion

In this work, we demonstrate that the zero-shot performance of frozen TSFMs can be significantly improved without parameter updates by mitigating context poisoning. We introduce GITCO, a deployable Gate-Router-Critic pipeline that identifies and suppresses structurally disruptive input patches at inference time. Evaluated on TimesFM 2.5, GITCO recovers a mean $+1.95\%$ MASE reduction across 53 datasets, capturing $89.9\%$ of the theoretically achievable improvement ceiling under the defined probe vocabulary.

The question of which architectural properties (e.g., attention design, patch stride, pretraining corpus) dictate the learnability of $\Phi_M$ constitutes the natural next stage of this research. Ultimately, these findings establish that context improvability is a joint property of the time series and the architecture, positioning inference-time input optimization as a persistent and scalable axis for foundation model enhancement.

## 6. Limitations

GITCO is evaluated on 53 GIFT-Eval datasets and two frozen TSFMs, with the strongest positive result observed on TimesFM 2.5. The Chronos2 result shows that context improvability does not guarantee a learnable or deployable gate under the same meta-feature vocabulary, so architecture-specific validation remains necessary. In addition, the reported oracle and Captured Improvement Ratio are defined only with respect to the fixed set of three Critics and the SMA-based denoising operator; richer intervention spaces may change both the attainable improvement and the fraction captured. Future work should evaluate larger model and dataset collections, alternative intervention operators, and robustness under distribution shift.

## Impact Statement

This paper presents work whose goal is to advance the reliability and safety of Time Series Foundation Models (TSFMs). By introducing a precision-first gating mechanism that prevents catastrophic forecasting errors caused by context poisoning, our framework contributes to the more robust deployment of predictive models in critical real-world systems (e.g., energy forecasting, resource allocation). There are many potential societal consequences of advancing machine learning and forecasting capabilities, none of which we feel present acute ethical risks that must be specifically highlighted here.

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

## A. Appendix

*Table 3.* Evaluation Dataset Summary

| Frequency Band | Example Datasets | TimesFM 2.5 | Chronos2 |
|---|---|---|---|
| Sub-hourly | `LOOP_SEATTLE/5T`, `SZ_TAXI/15T` | ✓ | ✓ |
| Hourly | `LOOP_SEATTLE/H`, `M_DENSE/H` | ✓ | ✓ |
| Daily | `M_DENSE/D`, `ETTh1`, `Weather/D` | ✓ | ✓ |
| Weekly / Monthly | `m4_monthly`, `us_births` | ✓ | ✓ |
| Other | Economics, retail variants | ✓ | ✓ |
| **Total** | | **53** | **53** |

## B. The Gating Primacy Principle

Across both models, the Gate is the dominant value-generating component of the GITCO system. We formalize this as:

**Gating Primacy Principle.** *Let $\mathcal{D}^+$ denote improvable datasets with mean improvement $\mu^+$, and $\mathcal{D}^-$ non-improvable datasets with mean degradation magnitude $|\mu^-|$ under erroneous intervention. Let Recall and FPR denote the Gate's true positive and false positive rates. Expected system value is:*

$$\mathbb{E}[\Delta\text{MASE}] = |\mathcal{D}^+| \cdot \text{Recall} \cdot \mu^+ \; - \; |\mathcal{D}^-| \cdot \text{FPR} \cdot |\mu^-| \tag{1}$$

*When $|\mu^-| > \mu^+$, minimizing FPR delivers strictly more expected system value than maximizing Recall.*

The condition $|\mu^-| > \mu^+$ holds empirically and is structurally motivated: denoising a genuinely improvable context produces bounded, incremental gains, while suppressing legitimate signal structure in a clean or periodic series can disproportionately degrade accuracy. This asymmetry is not a coincidence of dataset composition; it reflects the fundamental difference between a Critic operating in a well-posed regime (the Gate correctly identifies a poisoned context) and one operating in an ill-posed regime (the Gate incorrectly identifies a clean context as poisoned).

The TimesFM 2.5 results provide empirical validation of this principle under rigorous out-of-sample evaluation. The pipeline achieves a Captured Improvement Ratio (CIR) of 89.9%, capturing a cumulative $+57.33\%$ ($\Sigma\Delta\%$), despite operating at only 57.6% recall. By maintaining high precision (78.0%), the system successfully isolates high-leverage opportunities while avoiding the severe penalties of false alarms (where 83.3% of incorrect interventions degraded MASE). The Chronos2 results instantiate the principle negatively: because no gate decision boundary could be induced that met acceptable precision thresholds, the entire optimization pipeline collapsed under cross-validation. The practical implication is direct: the utility of inference-time context optimization is bounded entirely by gate calibration. Improvements to the Critic or Router are strictly additive; without a precision-safe Gate to shield the model from ill-posed interventions, the system cannot be deployed.

## C. Experimental Details

### C.1. Dataset Composition

Table 3 lists all 53 GIFT-Eval datasets used for both TimesFM 2.5 and Chronos-2 evaluation, grouped by temporal frequency band. GIFT-Eval was selected for its breadth across frequency, domain, and structural complexity, which is necessary to characterize context sensitivity profiles without overfitting to a single data regime.

### C.2. Sliding-Window Protocol

For a series of length $T$, context length $L$, and forecast horizon $H$, windows are extracted at stride 1:

$$\mathcal{W}_k = \big(x_{k:k+L}, \, x_{k+L:k+L+H}\big), \quad k = 0, 1, \ldots, \min(T - L - H, \, W_{\max} - 1), \tag{2}$$

with $W_{\max} = 300$. Per-dataset MASE is the mean across all extracted windows.

## C.3. Metric Definitions

The evaluation metrics are defined as follows:

$$\Delta\%_d = 100 \times \frac{\text{MASE}_d^{\text{base}} - \text{MASE}_d^{\text{GITCO}}}{\text{MASE}_d^{\text{base}}}, \tag{3}$$

$$\sum \Delta\% = \sum_{d:\, g_d=1} \Delta\%_d, \tag{4}$$

$$\text{Precision} = \frac{|\{d : g_d = 1,\ \Delta\%_d > 0\}|}{|\{d : g_d = 1\}|}, \tag{5}$$

$$\text{CIR} = \frac{\sum_d \Delta\%_d}{\sum_d \Delta\%_d^{\text{oracle}}}. \tag{6}$$

For non-intervened datasets, $\Delta\%_d = 0$ by construction. The oracle improvement, $\Delta\%_d^{\text{oracle}} = \max(0, \max_r \Delta\%_{d,r})$, represents the maximum accuracy gain achievable on dataset $d$ across the fixed vocabulary of expert probes $r$. Oracle labels are bounded by this defined probe vocabulary and denoising operator; GITCO does not claim globally optimal intervention.

The Captured Improvement Ratio (CIR) quantifies the end-to-end efficiency of the pipeline's out-of-sample decision making. While metrics like classification accuracy treat all routing errors equally, CIR operates as a value-weighted metric. It measures the fraction of the theoretical improvement ceiling (the denominator) successfully recovered by the system's actual interventions (the numerator). A CIR of 1.0 implies perfect hindsight-equivalent optimization, 0.0 indicates no net gain over the baseline, and negative values denote net degradation. By accounting for the asymmetric costs of false positives and suboptimal probe selections, CIR provides the most accurate representation of the system's practical deployment value.

