# OpenReview forum: "GITCO: Gated Inference-Time Context Optimization in TSFMs"
_ICML.cc/2026/Workshop/FMSD — FMSD @ ICML 2026 Poster_

### Official Review · Reviewer_g2WV · 2026-05-15
**Interesting inference-time optimization idea, but limited evidence and under-specified methodology**

**Rating:** 6
**Confidence:** 4

**Review:**

This paper proposes GITCO, a gated inference-time context optimization framework for patch-based time-series foundation models. The method uses a Gate to decide whether an input context should be modified, a Router to select among different probe Critics, and a Critic to identify a disruptive patch that is then smoothed with a 5-point simple moving average before forecasting with a frozen TSFM. The paper frames this as a way to mitigate context poisoning, where anomalous or misleading patches in the input history degrade zero-shot forecasting performance.

The main strength of the paper is that the problem is relevant and practical, and the paper is easy to follow along. Improving frozen TSFMs at inference time without updating model weights is an interesting direction. The Gate-Router-Critic decomposition is intuitive, and the precision-first gating idea is sensible because intervening on clean time series can easily destroy useful signal. The component ablation also supports the importance of gating, i.e., always intervening performs poorly, while the full Gate + Router system performs best among the reported variants. The paper is clearly motivated and fits the workshop theme well.

However, the empirical evidence is not yet fully convincing. The main positive result is on TimesFM 2.5, with a mean MASE reduction of only 1.95% across all 53 datasets and 4.30% on the intervened subset. This may be useful if the method is cheap and safe, but the gain is modest and should be supported with stronger statistical analysis, such as confidence intervals, significance tests, or per-dataset improvement distributions. The paper should emphasize the actual MASE gain at least as much as the CIR number.

A major limitation is generality. The method appears to work mainly for TimesFM 2.5. For Chronos2, the same protocol does not produce a deployable classifier, which suggests that the proposed meta-feature-based gate may not transfer reliably across TSFM architectures. The paper presents this as evidence for architecture-specific context sensitivity profiles, which is an interesting idea, but it also weakens the broader claim that GITCO is a general inference-time optimization method for TSFMs. The conclusions should be toned down to make clear that the strongest evidence is currently architecture-specific.

The method is also under-specified in important places. The paper does not provide enough detail about the training procedure for the Gate, Router, and Critics, the exact meta-feature set, the MLP architecture used by the Critics, the construction of oracle labels, or the relevant hyperparameters. Since the method depends heavily on learned gating and patch selection, these details are necessary for reproducibility. The Router accuracy is also reported as roughly random for a three-class problem, and the justification that the Router functions as a regret minimizer would be more convincing if the paper showed the distribution of probe regrets or how often the selected probe is close to the oracle-best probe.

I would also like to see stronger baselines. The ablation compares GITCO to internal variants such as always intervene, Gate-only, and Router-only, but it does not compare against simple external heuristics such as random patch smoothing, highest-variance patch smoothing, z-score outlier correction, volatility-based smoothing, or standard robust preprocessing. Without these baselines, it is difficult to know whether the learned Critic and Router are substantially better than simpler anomaly-denoising methods.

Overall, I find the idea promising and relevant for the workshop, but the evidence is still preliminary. The paper presents a useful direction for inference-time input optimization in TSFMs, but the modest gains, limited success beyond TimesFM 2.5, lack of strong heuristic baselines, and incomplete methodological details prevent a stronger recommendation. I recommend marginal acceptance, mainly because the idea is interesting and workshop-appropriate, but the claims should be made more conservative and the method should be described more fully.

---

### Official Review · Reviewer_MGMa · 2026-05-20
**Lightweight inference-time patch filtering for TSFMs**

**Rating:** 6
**Confidence:** 3

**Review:**

## Summary

GITCO is a three-component inference-time wrapper (Gate, Router, Critic) for frozen patch-based TSFMs. Anomalous patches can disproportionately capture attention and degrade zero-shot forecasts ("context poisoning"). Without weight updates, a meta-feature-driven Gate decides whether to intervene, a Router selects among three expert Critics, and the most disruptive patch is suppressed via 5-point SMA. Evaluated on TimesFM 2.5 across 53 GIFT-Eval datasets under K=11-fold CV, GITCO achieves +1.95% mean MASE reduction, capturing 89.9% of the oracle improvement ceiling. A negative result for Chronos2 is also reported via the new concept of *context sensitivity profiles*.

## Strengths

- Practically relevant framing; weight-frozen inference-time optimization is directly useful for production deployments
- The Chronos2 negative result is honest and insightful, distinguishing between oracle improvability existing and being learnable from a given feature vocabulary is a genuinely useful contribution
- Rigorous evaluation: K=11-fold CV, out-of-sample gating, and the value-weighted CIR metric are all well-motivated

## Areas for Improvement

- The +1.95% MASE improvement is reported only in relative terms with no baseline MASE values, making practical significance hard to assess
- Router CV accuracy of 33.3% ± 28.4% is barely above chance; the "flat improvement landscape" justification is plausible but undemonstrated
- No simple baselines (e.g., global SMA, highest-variance patch removal) are included, leaving it unclear how much the learned pipeline adds over naive data cleaning

## Detailed Comments

- Adding one or two naive data-cleaning baselines would significantly clarify the contribution of the learned components
- Given the Router's near-random accuracy, it is worth asking whether a single universal Critic would suffice and simplify the pipeline
- Baseline MASE values per dataset (or at least aggregate) should be reported to contextualize the improvement magnitude

## Justification of Score

GITCO addresses a real problem with a clean design and honest reporting. The negative Chronos2 result and the precision-first gating principle are the paper's most interesting contributions. However, the improvement magnitude is small and hard to contextualize, the Router is nearly non-functional by its own metrics, and the absence of simple baselines leaves genuine uncertainty about how much the learned pipeline contributes.

---

### Official Review · Reviewer_UB8z · 2026-05-21
**Review of Gated Inference-Time Context Optimization (GITCO)**

**Rating:** 8
**Confidence:** 4

**Review:**

## Summary

This paper introduces **GITCO** (Gated Inference-Time Context Optimization), a lightweight inference-time framework for improving zero-shot forecasting accuracy in patch-based Time Series Foundation Models (TSFMs) by remedying context poisoning. The core contribution is a three-component pipeline: Gate, Router, and Critic. This inference-time pipeline identifies and suppresses structurally disruptive input patches without modifying model weights. The paper also introduces *context sensitivity profiles* ($\Phi_M$) as a characterizable, model-specific property of TSFMs. Evaluated on TimesFM 2.5 across 53 GIFT-Eval datasets under $K=11$ fold cross-validation, GITCO achieves a mean MASE reduction of $+1.95\%$ across all datasets, capturing $89.9\%$ of the theoretically achievable improvement ceiling (CIR = 0.899).

## Strengths

### 1. Problem framing
The paper clearly identifies and names *context poisoning* as a concrete failure mode in attention-based TSFMs, and provides empirical grounding: over 50% of series across 53 datasets show marginal improvability, and 22 datasets benefit meaningfully from patch-level intervention. This is a useful empirical observation in its own right.

### 2. Asymmetric Gate design
The precision-over-recall operating point is rigorously motivated by the asymmetric loss condition $|\mu^-| > \mu^+$, where false-positive interventions on clean signals incur larger accuracy penalties than the gains from true-positive interventions on poisoned contexts. This is a principled design choice.

### 3. Lightweight and model-agnostic wrapper design
The Gate-Router-Critic pipeline requires no access to model internals and potentially adds negligible compute overhead. The modular design is clean and the component ablation in Table 2 effectively demonstrates the symbiotic necessity of each component.

## Weaknesses

### 1. Router accuracy is near-random
The 3-class routing accuracy of $33.3 \pm 28.4\$ is statistically indistinguishable from random performance, and the high variance across folds suggests the routing boundaries are unstable. The paper reframes this as "regret minimization", arguing that the improvement landscape is flat enough that precise probe selection is not critical. But this could also imply the Router lacks a coherent learned signal and its contribution may be largely coincidental.

### 2. Denoising operator and hyperparameter
The SMA with window $w=5$ is applied as an unargued heuristic. Why $w=5$ specifically? The paper states that downstream improvements are dominated by patch localization rather than operator complexity, but this claim rests on "preliminary ablations" that are not reported. A more thorough ablation over window sizes and alternative smoothing operators (e.g., exponential smoothing, median filter) would strengthen this claim considerably.

### 3. General-purpose vocabulary assumption is unvalidated and potentially self-defeating
GITCO uses a shared meta-feature vocabulary $\{\text{lumpiness, spectral entropy, CV, permutation entropy, ...}\}$ across both models, framing this as enabling cross-architecture comparability. However, the Chronos2 failure directly falsifies this assumption. $\Phi_{\text{Chronos2}}$ either requires features outside the span of this vocabulary, or has a diffuse boundary that resists compact characterisation at $N=53$. This raises a deeper unresolved tension:

- **Model-specific vocabularies** would undermine the framework's claimed generality, requiring re-engineering of the Gate for each new TSFM.
- **Expanding the vocabulary** risks severe overfitting at $N=53$ due to the curse of dimensionality in feature search.

There is also a subtle circularity risk: the chosen vocabulary likely reflects intuitions shaped by TimesFM-like architectures, which may partly explain why it succeeds on TimesFM 2.5 but fails on Chronos2. The paper does not adequately address whether a truly general, model-agnostic GITCO is achievable, or whether the framework is architecture-specific.

## Questions for the Authors

1. Can you provide the preliminary SMA ablation results referenced in Section 3.3, or a sensitivity analysis over window sizes $w \in \{3, 5, 7, 11\}$?
2. Is the shared meta-feature vocabulary a design choice or an empirical finding? Were alternative feature sets considered for Chronos2 specifically?
3. Could the Router be replaced by always selecting the best single probe (determined on the training folds), and would that match or exceed the current Router performance?